Larval settlement and metamorphosis in a marine gastropod in response to multiple conspecific cues

Cahill Abigail E. abigail.cahill@imbe.fr 1 2
Koury Spencer A. 1
1 Department of Ecology and Evolution, State University of New York at Stony Brook , Stony Brook , NY , United States
2 Institut Méditerranéen de Biodiversité et d’Ecologie marine et continentale (IMBE), Aix Marseille Université, CNRS, IRD, Avignon Université , Marseille , France
Pawlik Joseph
Electronic publication date: 2016 Jul 28
Publication date: 2016
Volume: 4
Electronic Location ID: e2295
Received 2016 May 2; Accepted 2016 Jul 6
Copyright: ©2016 Cahill and Koury
Copyright year: 2016
Copyright holder: Cahill and Koury
License: This is an open access article distributed under the terms of the Creative Commons Attribution License, which permits unrestricted use, distribution, reproduction and adaptation in any medium and for any purpose provided that it is properly attributed. For attribution, the original author(s), title, publication source (PeerJ) and either DOI or URL of the article must be cited.
License URL: https://creativecommons.org/licenses/by/4.0/

Keywords: Larva, Gastropod, Metamorphosis, Chemical cue, Selective settlement

Funding: The authors received no funding for this work.

==============================
Larvae of the marine gastropod Crepidula fornicata must complete a transition from the plankton, where they are highly dispersed, to an aggregated group of benthic adults. Previous research has shown that selective settlement of larvae on conspecific adults is mediated by a water-borne chemical cue. However, variable experimental conditions have been used to study this cue, and standardization is needed in order to investigate factors that may have weak effects on settlement. In this study, we developed a time-course bioassay based on a full-factorial design with temporal blocking and statistical analysis of larval settlement rates in the lab. We tested this bioassay by examining settlement in the presence of an abiotic cue (KCl), and biotic cues (water conditioned with adult conspecifics and conspecific pedal mucus). Results confirmed settlement in the presence of both KCl and adult-conditioned water, and discovered the induction of settlement by pedal mucus. This optimized, standardized bioassay will be used in future experiments to characterize the complex process of larval settlement in C. fornicata, particularly to measure components of potentially small effect.

Introduction

An important challenge in ecology is understanding how broadly-dispersed propagules locate sites where they will survive as adults. Many marine invertebrates with complex life cycles have vastly different distributions in their adult and larval stages. Planktonic larvae can disperse tens to hundreds of kilometers during their larval period, while the benthic adults are often sedentary and aggregated in spatially restricted habitats (Cowen & Sponaugle, 2009). Many larvae exhibit selective settlement, where the planktonic-benthic transition is mediated by physiological responses of larvae to physical or chemical cues associated with suitable sites for adults (Krug & Manzi, 1999) or from the adults themselves (Zimmer-Faust & Tamburri, 1994).

The terminology used to describe the process of transitioning from a larva to a juvenile is variable. Terms here are consistent with Pawlik (1992), where settlement refers to the entire process of transitioning from a planktonic larva to a benthic juvenile, while metamorphosis is used to indicate irreversible developmental changes that prevent a larva from returning to its previous planktonic lifestyle. Therefore, the developmental process of metamorphosis is contained within the ecological process of settlement.

Larvae may use physical or chemical cues to settle selectively in appropriate habitats, avoid inappropriate habitats, or delay metamorphosis until appropriate cues are sensed (Thorson, 1950; Woodin, 1986; Pechenik & Eyster, 1989; Morello & Yund, 2016). Environmental cues known to cause larval settlement are diverse and may be associated with biofilms (the gastropod Crepidula onyx, Zhao & Qian, 2002), organisms that provide food or habitat (the hydroid Proboscidactyla flavicirrata, Donaldson, 1974; the soft coral Alcyonium siderium, Sebens, 1983), conspecifics (the barnacle Semibalanus balanoides, Gabbott & Larman, 1987; the polychaete worm Phragmatopoma californica, Jensen & Morse, 1984; the oyster Crassostrea virginica, Zimmer-Faust & Tamburri, 1994), or avoidance of organisms with negative impacts (the polychaete worm Pseudopolydora kempi, Woodin, 1985).

The larvae of the calyptraeid gastropod Crepidula fornicata (Linnaeus, 1758) have a planktonic period of two to four weeks (Collin, 2003), which allows for long larval dispersal distances. Within a single location larvae also have a wide spatial distribution, demonstrated by plankton tows of a single estuary that found larvae present in areas without adults (Rigal et al., 2010). In contrast, the sedentary adults are patchily distributed within intertidal and shallow subtidal habitats (Henry, Collin & Perry, 2010; Hoch & Cahill, 2012). Within these habitats, adults exhibit a clumped distribution due to their tendency to form large, semi-permanent mating groups called stacks (Collin, 1995). The presence of small juveniles aggregated on adults demonstrates recruitment to these stacks (McGee & Targett, 1989; Cahill, 2015).

Selective settlement of C. fornicata larvae in response to cues has been studied in the field. Juveniles preferentially aggregate on adults and increased adult density increases larval recruitment (McGee & Targett, 1989; Bohn, Richardson & Jenkins, 2013a; Bohn, Richardson & Jenkins, 2013b; Cahill, 2015), consistent with the idea of a cue associated with conspecific adults. Additional support for the role of settlement cues comes from laboratory studies. Larvae metamorphose in response to dissolved or suspended cues, including increased concentrations of potassium chloride (KCl; Pechenik & Heyman, 1987), dibromomethane from coralline algae (Taris et al., 2010), and seawater conditioned with conspecific adults (Pechenik & Heyman, 1987; Pechenik & Gee, 1993; Bohn, Richardson & Jenkins, 2013b). This is also consistent with a conspecific waterborne cue, though the nature of the cue remains unknown. In some studies of settlement in C. fornicata, adult-conditioned seawater was prepared in the same vessel in which larvae were tested (Penniman, Doall & Pires, 2013), confounding the effect of the adult-conditioned water with any potential effect of pedal mucus produced by adult snails. Molluscan pedal mucus is known to affect settlement rates in other marine invertebrate larvae such as barnacles (Johnson & Strathmann, 1989; Holmes, 2002).

The existence of a conspecific waterborne cue that induces settlement in C. fornicata has been supported using a variety of experimental designs and assay conditions (e.g., Pechenik & Heyman, 1987; Pechenik & Gee, 1993; Bohn, Richardson & Jenkins, 2013b), but the chemical nature and biological mechanism of this cue remain unknown. Further characterization of the cue requires repeatable assays that control for biological, technical, and statistical sources of variation in measurements of the settlement process. Biological variability is due to larval age, genetic differences among larvae, and differences among egg masses in survival, growth, and development (Hilbish et al., 1999). Technical variability stems from protocols that differ in biotic and abiotic conditions known to affect larval settlement, including the mass of adults used to produce the cue or the density of larvae in a trial (Pechenik & Heyman, 1987; Pechenik & Gee, 1993; Padilla et al., 2014). An additional, statistical source of variability is introduced by analysis of settlement based on data collected at a single time point. Using this procedure, the results are sensitive to the form of the mathematical function assumed for settlement rate and dependent on which time point is selected (e.g., performing an ANOVA on the proportion of larvae settled at a single timepoint assumes a constant rate of settlement through time).

Following the observation that C. fornicata distributions differ between the planktonic larval and benthic adult life stages, and building on previous work with conspecific cues, we aimed to provide a standard by which we can then investigate the mechanism responsible for this ecological transition. We developed a bioassay using a fully factorial, randomized block design, which accounted for biological variability. Technical variability was minimized by optimization and control of biotic and abiotic experimental conditions. To address the problem of statistical variability, we introduced an analysis that estimated settlement rates as a single parameter by fitting an exponential function using multiple observations from regular intervals. Measuring settlement through time with rates also allows for differentiation among cues of different induction strength and response times. This analysis improved the ability of experiments to repeatably detect changes in settlement rates in the complex larval settlement system of C. fornicata.

Methods

Crepidula fornicata collection and husbandry

We collected adult Crepidula fornicata from Crab Meadow Beach (Northport, New York, USA: 40°55′46′′N, 73°19′8′′W) in July 2013 and returned them to the lab the same day. No permit is required for collecting this species in New York State, and animals were not returned to the field. Adult females were removed from their substrates to check for incubating egg capsules. Capsules with larvae that were ready to hatch (stage IV veligers sensu Leroy et al., 2013) were selected and hatched by physically agitating them in a bowl of filtered seawater at room temperature. Larvae from three females were combined for rearing in cultures of 800 ml of 1 µm-filtered seawater (FSW) at a concentration of one larva per four ml. We collected seawater from an underground well at Flax Pond Marine Laboratories, Old Field, New York, USA (40°57′49′′N, 73°08′26′′W). We fed larvae 40,000 cells/ml of the alga Isochrysis galbana (clone T-ISO) daily. We maintained larval cultures at 20 ° C and replaced FSW via reverse filtration every three to four days. We tested for larval competence (ability to metamorphose) every two days once the larvae developed shell brims (Pechenik, 1984) and were at least 750 µm long (Pechenik & Heyman, 1987). Competence was tested by placing 12–24 larvae (1–2 larvae from each culture) in 20 mM KCl solution for 8 h. It is not possible to work with a larval culture that has reached 100% competence, because at this point many larvae have spontaneously metamorphosed and are no longer available for experiments. We therefore started the experiment within 24 h of the larvae reaching 75% competence (75% of larvae metamorphosed in response to KCl; Pechenik & Heyman, 1987), which occurred 19 days post-hatch.

Preparation of test solutions

We tested three factors for induction of larval settlement: seawater conditioned with conspecific adults, conspecific pedal mucus, and 20 mM elevated KCl. Prior to the start of the experiment, we acid-cleaned all glassware for ten minutes in 10% concentrated HCl, rinsed them in deionized water, and then autoclaved them.

To create adult-conditioned seawater (ACW), we placed 100 g of adult C. fornicata (shell and wet tissue mass) and one liter of FSW into a beaker, then oxygenated the water for twelve hours. Large epibionts (e.g., barnacles, macroalgae) were removed from the shells, but shells were not otherwise treated. One liter of FSW without C. fornicata, to be used as a control, was also oxygenated. After twelve hours, adults were removed and ACW and FSW were filtered to 40 µm with a Nitex mesh filter. Temperature, dissolved oxygen, salinity, and pH were measured at all preparation steps. The biotic and abiotic parameters for the bioassay (salinity, trial length, mass of adults used for cue preparation, etc.) are listed in Table 1. These values were chosen based on optimization studies for each parameter where settlement was measured over a range of values. The optimal value was chosen based on settlement rates. When multiple values gave similar settlement rates, we made our choice based on logistical concerns (details in Fig. S1).

Table 1 Optimized values of abiotic and biotic parameters in the bioassay.

Further details about the optimization process can be found in Fig. S1.

Variable	Range tested	Optimal value	
pH	7.9–10.2	8.0–8.3	
Salinity	25–40	27–30	
Adult mass used for ACW	4 g–800 g	100 g	
ACW preparation time	1 h–24 h	12 h	
Larvae per replicate 20 ml glass	5–20	10	
Trial length	8 h–212 h	48 h	
Sampling time intervals	4 h–24 h	12 h	

We prepared pedal mucus treatments (replicates with pedal mucus left by adult snails; PMG) at the same time that ACW was prepared. Individual replicates of the experiment were conducted in 60 ml drinking glasses (shot glasses or shooters; hereafter “glasses”). For each PMG replicate, a single small (∼15 mm) adult C. fornicata was added to all glasses for 12 h at the same time that the ACW was prepared. Glasses were filled with 35 ml FSW and covered to prevent evaporation and snail escape. All snails began the 12 h period at the bottom of the glass, and so even though some crawled above the water line during this period, they still left mucus footprints in the glasses. During the ACW preparation, glasses not receiving the PMG treatment had adults removed and were acid-cleaned and autoclaved to remove the mucus. To prevent desiccation of the mucus, we did not remove adults from glasses receiving the PMG treatment until immediately before the start of the experiment; these glasses were then drained.

For KCl treatments, we prepared a concentrated solution of 200 mM KCl in distilled water, which was stored until use in the experiments (following Pechenik & Heyman, 1987).

Bioassay

Because larvae are expected to become more likely to metamorphose as they develop, we used a randomized complete block design to account for larval age, using the start date of the experiment as a blocking factor (i.e., blocks were run through time). All blocks used the same batch of larvae and therefore larvae in the later blocks were older. All blocks were run within a ten-day period. All three factors (ACW, PMG, KCl) had two levels (present or absent) and were tested using a factorial design (eight possible treatments; Fig. 1). The treatment combination where all factors were absent corresponded to FSW and served as a negative control. Each treatment combination had three replicate glasses for 24 replicates per block (72 replicates total).

Figure 1 Experimental design.

The experiment contained three factors (potassium chloride, adult-conditioned water, and pedal mucus), each either present (+) or absent (−), crossed in a factorial design for a total of eight treatments. Each treatment had three replicates (grey lines) with ten larvae in each replicate. The star indicates the treatment where all factors were absent. This treatment was equivalent to filtered seawater and served as the control. This design was repeated in three blocks through time (see details in text).

Each glass in the bioassay contained 20 ml of the test solution (Fig. 1). To make the KCl treatments, we added 2 ml of the concentrated KCl solution to the test solution of the replicate (either FSW or ACW; Fig. 1) for a final KCl concentration of 20 mM elevated above FSW. Ten larvae were then individually pipetted into each glass. Larval growth and development in many marine larvae, including C. fornicata, varies among cultures (rearing beakers). This variation was accounted for by placing one larva from each rearing beaker (ten total beakers) into each replicate glass. The same set of rearing beakers was used in all three blocks of the experiment.

Every 12 h, we counted the number of larvae metamorphosed in each replicate, and recorded any mortality. Metamorphosis was measured by the loss of the velar lobes, meaning that it was an irreversible step in development. We removed metamorphosed juveniles and dead larvae from the trial at each time point. To limit bacterial growth and the buildup of waste products, after 24 h we replaced the test solution in the glasses with clean glasses containing new ACW, KCl, and PMG prepared as described above, and individually pipetted larvae into the clean glasses. The total time of the experiment was 48 h, which included five time points and two different preparations of test solutions. Larvae were not fed during the experiment.

Data analysis

We conducted an analysis based on settlement rates rather than the proportions of larvae settled at a fixed time point, since proportions are sensitive to the underlying mathematical function of settlement rate and the timepoint selected for the analysis. We modeled larval settlement by predicting the proportion of larvae settled (y) at each timepoint (t, in hours) using the cumulative distribution function for the single-parameter exponential model: y=1−e−λt.

Given a constant probability of settlement, the waiting times for a single individual to settle (ti) in a given treatment are exponentially distributed. Note that this model assumes that all larvae in the experiment are developmentally capable of settling (competent), although trials began when only 75% of larvae were competent; it was not possible to wait for 100% competence due to high rates of spontaneous settlement under these conditions (see above). Incomplete competence will not affect the overall results if competence is equal in all treatments, a reasonable assumption given our random assignment of larvae to treatments. The exponential distribution is defined by the single parameter λ, which can be estimated as λ ˆ=n∑i=1nti.

However, because not all larvae settled during the first 48 h, we calculated λ incorporating Type I censoring with the following equation: λ ˆ=r∑i=1rti+Tn−r,

where n is the total number of larvae tested and r is the number of larvae that settle during the bioassay. Thus, (n − r) is the number of non-metamorphosed larvae at time T which represents the end of the bioassay (fixed at 48 h for all blocks). By modeling larval settlement in this way, the settlement rate for each replicate could be summarized with a single value (λ ˆ) that used time-course data and also accounted for Type I censoring.

To analyze the experimental data, we corrected the number of larvae tested (n) for mortality (average mortality per block = 2%, or approximately five larvae; mortality was consistent among treatments) and then calculated λ ˆ for each replicate. We used data from the first block of the experiment to calculate the correlation between the predicted number of larvae settled for each replicate at each timepoint (based on λ) and the observed number of larvae settled.

The full experiment was then analyzed using λ ˆ as a response variable in an analysis of variance. Blocks were treated as random effects and experimental treatment factors (ACW, PMG, KCl) were treated as fixed effects. The treatment where all factors were absent was equivalent to FSW and served as a control (Fig. 1). Planned comparisons were conducted of each factor against this control (H0: KCl = FSW, ACW = FSW, and PMG = FSW). The significance of planned comparisons was assessed using the 95% confidence intervals of the mean difference between the treatments. Confidence intervals that did not overlap with zero indicated a significant difference between the treatment and the control (Motulsky, 2010).

Finally, in order to compare our results using rates to results that would be obtained by using proportions, we calculated an ANOVA on the arcsine-squareroot transformed proportions of larvae settled at each timepoint in the analysis (four separate ANOVAs). Statistics were conducted using JMPIN (Version 4.0.4, 2001; SAS Institute, Cary, NC, USA) and R 3.0.1 (R Core Team, 2013).

Results

Modeling settlement rates

The proportion of larvae expected to settle over successive twelve-hour intervals was predicted by modeling settlement with the rate parameter λ from the exponential distribution. The fit of the estimated values of λ (λ ˆ) to the observed cumulative proportions of larvae settled is illustrated with data from the first block of the experiment (Fig. 2). The correlation of predicted and observed values was high (overall r = 0.930, p < 0.001) and consistent across treatments (Fig. 2A). The slope of the best-fit line of predicted and observed data was less than one (slope = 0.810 ± 0.030 SE), indicating that the model slightly underpredicted at most time points (Fig. 2B). Modeling settlement as the rate parameter λ was a more informative statistical analysis than using proportions of larvae settled at a given time. Performing the analysis on arcsine-transformed proportions using ANOVAs yielded inconsistent results, such that the significance of both main effects and interaction terms depended on the time point selected for the analysis (Table 2, Table S1).

Figure 2 Fit of λ to observed data.

(A) Observed larval settlement (circles) and the values predicted by λ (lines) at each time point, plotted as the proportion of larvae settled at each point. Values calculated based on the first block of the experiment. Error bars represent standard error. There are close correlations between observed and expected values for all treatments: adult-conditioned water (ACW; r = 0.977, p < 0.001), adult-conditioned water and pedal mucus (ACW*PMG; r = 0.957, p < 0.001), pedal mucus glasses (PMG; r = 0.965, p < 0.001), and filtered seawater (FSW; r = 0.710, p < 0.001). (B) Plot of observed versus predicted proportions of larvae settled in each glass at all time points. Values calculated based on the first block of the experiment. The best-fit line is the linear regression to the data (y = 0.810x + 0.006; r = 0.930; p < 0.001).

Table 2 Heat map illustrating the change in significance through time of main effects and interactions tested, analyzed using ANOVAs on arcsine square root transformed proportions; calculations were done at each time step.

Values in cells represent p-values for each factor at each timestep. Dark grey: p < 0.001; light grey: 0.001 < p < 0.05; white: p > 0.05. Full ANOVA tables for each timestep can be found in Table S1.

Source of variation	12 h	24 h	36 h	48 h	
Adult Conditioned Water (ACW)	0.408	0.301	0.002	0.006	
Pedal Mucus Glass (PMG)	0.936	0.949	0.0514	0.058	
Potassium Chloride (KCl)	<0.001	<0.001	<0.001	<0.001	
PMG*ACW	0.750	0.223	0.046	0.026	
PMG*KCl	0.317	0.598	0.006	0.004	
ACW*KCl	<0.001	<0. 001	<0.001	<0.001	
ACW*PMG*KCl	0.207	0.182	0.145	0.096	

Larval settlement rates

All factors showed an increased settlement rate (λ) relative to the filtered seawater (FSW) control (Figs. 3 and 4). The linear model contained two statistically significant treatment effects (KCl, F1,62 = 19.43, p < 0.001; KCl*ACW, F1,62 = 14.57, p < 0.001; Table 3), with block effects through time accounting for 24.6% of total variation. The strength of the artificial cue was likely responsible for significant interaction effects (KCl*ACW, Table 3), as complete induction by KCl allowed for no additional effect of the ACW treatment.

Figure 3 Settlement rates of larvae in response to different experimental factors.

Time-course data of larval settlement in adult-conditioned water (ACW; red line), filtered seawater (FSW; grey line), potassium chloride (KCl; black line), and pedal mucus glasses (PMG; blue line). Points represent running averages across all three experimental blocks; error bars represent 1 SE.

Figure 4 Comparisons of settlement factors to the control.

The difference in settlement rate (λ, in units of ln(number larvae settled per hour)) between each settlement factor and the control for adult-conditioned water (ACW; red), potassium chloride (KCl; grey), and pedal mucus glasses (PMG; blue). Bar height represents the difference between each treatment mean and the filtered seawater control, and error bars represent the 95% confidence intervals on those differences. Error bars that do not overlap with zero indicate treatments with a significantly higher settlement rate than the control at α = 0.05.

Table 3 Analysis of variance table, conducted using rates (λ) as the response variable.

Significant effects at p = 0.05 are highlighted in bold.

Source of variation	df	SS	MS	F	p	
Block effect (time of experiment)	2	1.74E–03	8.71E–04	–	–	
Adult Conditioned Water (ACW)	1	6.71E–05	6.71E–05	1.29	0.26	
Pedal Mucus Glass (PMG)	1	1.26E–05	1.26E–05	0.24	0.63	
Potassium Chloride (KCl)	1	1.01E–03	1.01E–03	19.43	<0.001	
PMG*ACW	1	1.07E–04	1.07E–04	2.04	0.16	
PMG*KCl	1	1.26E–04	1.26E–04	2.43	0.12	
ACW*KCl	1	7.59E–04	7.59E–04	14.57	<0.001	
PMG*ACW*KCl	1	6.51E–06	6.51E–06	0.12	0.72	
Error	62	3.23E–03	5.21E–05	0.00		
Total	71	7.06E–03				

The mean settlement rates (λ, in units of ln(number larvae settled per hour)) for KCl, ACW, PMG, and FSW were 0.01796, 0.01218 0.007234, and 0.000715, respectively. The 95% confidence intervals of the difference in mean settlement rates between each treatment factor and the FSW control did not overlap with zero (difference in λ between KCl − FSW = 0.01075 − 0.02374; ACW − FSW = 0.005248 − 0.017572; PMG − FSW = 0.002583 − 0.010456). This indicates that all factors induced settlement in C. fornicata (Fig. 4).

Discussion

This experiment confirmed larval settlement induction in Crepidula fornicata by 20 mM elevated potassium chloride (KCl) as well as adult-conditioned water (ACW) and discovered the inductive effect of pedal mucus (PMG) in the absence of ACW (Figs. 3 and 4). Induction of settlement in C. fornicata by KCl and ACW has been previously reported (Pechenik & Heyman, 1987; Pechenik & Gee, 1993; Bohn, Richardson & Jenkins, 2013b). We reproduced these previous results with statistical significance, validating the time-course bioassay and analysis using the rate parameter λ.

Using λ to measure settlement rather than proportions of larvae settled at a single timepoint allowed for a consistent analysis. When using proportions, the significance of both main effects and interactions varied depending on the timepoint selected (Table 2), making the analysis less robust to variation in experimental duration. The use of λ will be particularly important when measuring potentially weak effects of induction, which induce settlement at a slower rate. The experiment presented here does not allow for any characterization of the chemical cues involved in this complex settlement system. However, the bioassay and analysis now provide a standardized protocol and statistical analysis for the estimation of subtle differences in settlement rates under various treatments.

Pedal mucus is a weak inducer of settlement which was not detectable using the proportions of larvae settled at particular timepoints (Table 2, Table S1), but which we were able to detect by analyzing settlement rates (Figs. 3 and 4). Previous work has shown that molluscan pedal mucus affects settlement in other organisms (the barnacles Balanus glandula and Semibalanus balanoides: Johnson & Strathmann, 1989; Holmes, 2002). Additionally, a study of settlement in C. fornicata used a combined treatment of adult-conditioned water and pedal mucus to induce settlement (Penniman, Doall & Pires, 2013). However, our study is the first to demonstrate the inductive effect of mucus in the absence of adult-conditioned water.

There may be other weak inducers of settlement that can be detected with the analysis of settlement rates. Bacteria and biofilms have often been implicated in larval settlement, including in Crepidula species (C. onyx; Zhao & Qian, 2002). To limit bacterial growth in the current experiment, larvae were transferred into new water and glasses after 24 h. There was no accelerated settlement in time points immediately preceding transfers, indicating that any effect of bacterial populations on settlement is overwhelmed by the signal of the conspecific cues in ACW and PMG treatments. However, we did not test explicitly for the effect of biofilms and bacteria, which have been shown to be a weaker effect than conspecific cue in C. onyx (Zhao & Qian, 2002). The bioassay can be used to test for this effect, as well as measuring its strength relative to conspecific cues in C. fornicata.

The sensitivity of the bioassay and analysis also allows for the characterization of the chemical cues involved in C. fornicata settlement. Many chemical inducers of settlement are known from other gastropods, including carbohydrates (Alderia modesta, Krug & Manzi, 1999), metabolites (Phestilla sibogae, Hadfield & Pennington, 1990), volatile halogenated organic compounds (Haliotis discus hannai, Kang, Kim & Kim, 2004), and peptides (Adalaria proxima, Lambert, Todd & Hardege, 1997). Work is currently underway to use the bioassay to begin to characterize the cue present in both ACW and PMG (A Cahill & S Koury, 2014, unpublished data).

The optimization and standardization of the conditions of the bioassay allowed us to account for several sources of variation in larval settlement. Settlement rates in this study were variable among temporal blocks, with nearly one quarter of the variation in settlement rate explained by block effects. Larval settlement rate is expected to change through time as larvae become competent during development. The use of a blocked design allowed us to statistically account for this variation while increasing our sample size beyond the number of larvae that could be tested at one time.

Differences in competency among larvae within a block could be explained by differences in larval growth rate (Pechenik & Lima, 1984; Pechenik, Estrella & Hammer, 1996) due to food availability or temperature (Padilla et al., 2014). However, we controlled variation by rearing all larvae on the same diet and at the same temperature. Variation in larval growth rate of C. fornicata is also influenced by sire (Le Cam et al., 2009) and maternal effects (Hilbish et al., 1999). Another potential source of variation among broods was due to the fact that we artificially hatched the larvae, rather than waiting for natural hatching. We attempted to control for this by only hatching very late-stage embryos, but some broods may have been more developmentally advanced than others, leading to higher settlement rates. Due to the size of the experiment and logistical constraints, we were unable to statistically account for these effects (i.e., by blocking according to brood). By randomly mixing larvae from multiple broods, we spread unknown brood-related variation evenly among all treatments and blocks, so such variation does not impact our overall results regarding the different treatments.

Variation in settlement rate among larvae may also explain the discrepancy between the fact that 75% of the larvae tested in KCl were competent before an experiment began, but that settlement was consistently lower than 75% in the KCl treatments under experimental conditions (Fig. 3, Fig. S1E). The KCl used for the competency tests was prepared in the same way as that for the experiment itself. Although the use of KCl solution prepared in DI water reduced the salinity to approximately 26, slightly below values reported as optimal in Table 1, these treatments nonetheless showed high levels of settlement, and larval behavior did not appear impacted. The salinity levels remained near or above the values observed at the collection site in Northport (e.g., salinity in June 2015 was 24.1; A Cahill, 2015, unpublished data).

In addition to variation among larvae, there was variation among preparations of both the ACW and the PMG: different adult animals were used for each preparation. The standardization of the abiotic and biotic parameters of the experiment minimized differences among blocks associated with preparation of adult-conditioned water (Table 1, Fig. S1). However, in the absence of a clearly identified chemical that induces settlement, it remains impossible to control the exact concentration of cue delivered to the larvae.

Food limitation has also been shown to increase settlement rates in C. fornicata (Pechenik, Estrella & Hammer, 1996). This may have played a role in our experiment, since larvae were not fed during the settlement trials. This food limitation was the same across treatments, and does not explain the high settlement rates in KCl and ACW treatments relative to FSW. It may, however, explain the small number of larvae that settled in the FSW treatment despite the absence of settlement cues (Figs. 3 and 4). This spontaneous, background settlement potentially due to food limitation should be the same across treatments.

Conclusion

We developed an optimized time-course bioassay to estimate larval settlement rates in C. fornicata. We replicated previous results by demonstrating settlement in response to both elevated concentrations of KCl and a waterborne conspecific cue. For the first time, we demonstrated that pedal mucus from adult conspecifics induces settlement in the absence of adult-conditioned water. Future work using our new bioassay will characterize these cues and investigate other potentially weak inducers of settlement.

Supplemental Information

Table S1 Supplemental table: settlement data analyzed using proportions

Click here for additional data file.

Figure S1 Supplemental figure: optimization of experimental parameters

Click here for additional data file.

Supplemental Information 1 Readme file explaining the supplemental data for this manuscript

Click here for additional data file.

Supplemental Information 4 Raw settlement data for the manuscript

Click here for additional data file.

Supplemental Information 5 Data on settlement rates used to produce Fig. 2A

Click here for additional data file.

Supplemental Information 6 Settlement data used to produce Fig. 2B

Click here for additional data file.

Supplemental Information 7 Settlement data used to produce Fig. 3

Click here for additional data file.

Supplemental Information 8 Settlement rate data used to create Fig. 4

Click here for additional data file.

W Bruno, A Pellman-Isaacs, and J Tay helped with larval rearing. We thank J Levinton and W Eanes for lab space and equipment, J Rohlf for statistical advice, and J Levinton, D Padilla, D Futuyma, F Viard, W Eanes, J Rollins, E Rollinson, and several anonymous reviewers for comments on previous versions of the manuscript. This is contribution #1246 of the Department of Ecology and Evolution, Stony Brook University.

Additional Information and Declarations

Competing Interests

Author Contributions

Data Availability

The authors declare there are no competing interests.

Abigail E. Cahill and Spencer A. Koury conceived and designed the experiments, performed the experiments, analyzed the data, contributed reagents/materials/analysis tools, wrote the paper, prepared figures and/or tables, reviewed drafts of the paper.

The following information was supplied regarding data availability:

The raw data has been supplied as a Supplemental Information.

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
