# Peer review of "Larval settlement and metamorphosis in a marine gastropod in response to multiple conspecific cues"

_PeerJ, doi:10.7717/peerj.2295_

## Round 0.1 · original submission · Major Revisions

I now have two thoughtful reviews of your ms., both of which recommend revision before publication. Reviewer 1 has relatively minor comments that should be considered, including the use of the term "settlement" vs. "metamorphosis" and some methodological issues. Reviewer 2 has more critical issues with the methods and the validity of the results. Please address each point of each reviewer in your rebuttal letter and revise your ms. accordingly.

Reviewer 1 ·

Basic reporting

No Comments, paper seems fine and appropriate.

Experimental design

Manuscript meets the stated standards. Rationale is clear, methodology seems generally complete and reasonable. Just one question here: Why was the KCl solution prepared using distilled water? When that concentrated solution is added to seawater, the total salt concentration will decrease (by about 10%, with 2 ml of concentrated KCl solution added to 20 ml of seawater). Also, note that normal seawater already has KCl in solution, so that total concentration is more than just from the KCl that is added. Authors should address this issue, and show that it is not a problem, and also indicate what the final KCl concentration actually was, or at least what the final salinity was.

Validity of the findings

Validity seems fine.

Additional comments

Nice idea to standardize approach for determining factors inducing metamorphosis. You might want to reconsider use of the term "settlement". As used here, it means settlement followed by metamorphosis. But generally it just means settling out of the plankton, something that will happen if you just add formalin! So it could be confusing to casual readers. What you are really interested in is what triggers metamorphosis, so why not just talk about that?
Line 124: Use all caps: T-ISO
Line 129: Reword. "It is not possible...with a culture in which all larvae are competent to metamorphosis, because..."
Line 139: I think that most authors use ACSW for adult-conditioned seawater.
Line 281: indicate species
Table 3: After "Block Effect" I would add "(Age of larvae)"
Fig 1. It can't hurt to define ACW, PMG. Actually, why use these acronyms at all here in the caption, since the axes are labelled with the actual words?
Fig 3. The author does not really mean "Settement rates of different factors"! The factors are not settling! Also, indicate the acronyms in the text of the caption. Or label the symbols with real words rather than acronyms (plenty of room on the figure to do this).
Fig. 4. Indicate units for "settlement rate"; what are Y-axis units?

Reviewer 2 ·

Basic reporting

No comments

Experimental design

The authors pursue the laudable goal of designing a standardized bioassay for measuring effects of natural and artificial cues on settlement and metamorphosis of the gastropod, Crepidula fornicata. Many important details of culture methods and of the preparation of cues are described, more thoroughly than in most publications in this field. However, other important information is omitted. For example, it is stated (line127) that larvae were at least 750 um when used in experiments, but no other size information is given other than this minimum bound. Just as importantly, no age information is given at all. Given that these are field-collected, brooded larvae, one cannot know the fertilization date. However, it would have been better to wait for natural hatching and release of larvae from brooding females, rather than to artificially (mechanically) hatch egg capsules that appeared to be "near hatching" (line 119). The pooling of egg capsules of uncertain developmental status from three females may have contributed to the authors' difficulty in raising batches of larvae that achieved high levels of competence without spontaneous metamorphosis (line 129). Given the known existence of maternal and sire effects (cited by the authors) it would have been better not to mix broods, but perhaps to culture each brood separately until maximally competent, and then block the experiment on brood. Settlement frequencies in this study were low overall (Fig. 3) compared to other studies of effects of adult cue and KCl cited by the authors.

Validity of the findings

An important element of the experimental design is the use of settlement frequency (proportion) data, collected at several time points, to model a settlement function defined by a settlement rate (lambda). Lambda is then used as the response variable to compare treatment effects. The justification given is that (line 192) "proportions are sensitive to the underlying mathematical function of the settlement rate and the timepoint selected for the analysis." Later in the Results section, the authors claim (line 243), "Modeling settlement as the rate parameter lambda was a more informative statistical analysis than using proportions of larvae settled at a given time. Performing the analysis on arcsine-transformed proportions using ANOVAs yielded inconsistent results, such that the significance of both main effects and interaction terms depended on the time point selected for the analysis (Table 2, Table A1)." I believe that this conclusion is unfounded, and that modeling settlement rates as the authors have done does not usefully contribute to our understanding of larval settlement. In part this is a methodological/experimental design issue, but I'm discussing it under "Validity of the Findings" because this way of looking at larval settlement is purported to be the main contribution of the paper. My first objection to this approach is a priori: If you have collected data on a real biological outcome (frequency of settlement), then the analysis should work directly with the data that describe that outcome. The "inconsistent results" of the ANOVAs on transformed proportions that "depended on the time point selected for the analysis" probably reflect an interesting biological reality about the mechanisms of action of settlement cues. It does no good to use a real response variable (proportion or frequency of settlement) to estimate a parameter for a hypothetical settlement rate function that itself makes a physiologically unrealistic assumption (constant probability of settlement, line 197), and then look for treatment effects on that parameter. And despite the high r values claimed for the model (Fig. 2A), it is clear from that same figure that the settlement rate function does not always accurately predict settlement (e.g., data points for ACW*PMG). My second objection is that the authors' own ANOVAs on proportion data (Table 2) are actually more informative and sensitive than the ANOVA on settlement rate (Table 3). The settlement rate analysis in Table 3 cannot even detect a significant main treatment effect of adult-conditioned water, which was apparent in the ANOVA on proportions and in several previous studies cited by the authors.
The authors use an alternative to ANOVA, confidence interval comparison, to demonstrate a significant treatment effect of pedal mucus on settlement (Fig. 4). This effect had not emerged from the ANOVA on settlement rate, but there was a near-significant effect of mucus in the ANOVA on proportions. The report of settlement in response to adult pedal mucus is in fact a valuable contribution of this study. (By the way, the bars in Fig. 4 seem to be consistent with the source data in the supplementary files for that figure, but I can't reconcile them with the values reported in Results line 258). However, I wonder if the effect of pedal mucus could have been equally well demonstrated with a confidence-interval approach applied to proportion data from later time points in the bioassay, or even from an ANOVA like Table 2 in more homogeneous cohorts of larvae.

Additional comments

We need well-controlled, thoughtfully-designed and statistically-tractable bioassays in order to make any progress characterizing settlement cues in C fornicata and other animals. Your assay design (with the exception of the artificial hatching and pooling of egg masses) accomplishes that, but to my mind you haven't made a convincing case for using lambda instead of settlement frequencies as the response variable.

---

## Round 0.2 · accepted · Accept

The authors have done an excellent job of dealing with the thoughtful comments of the two reviewers. In particular, the reasons for the choice of statistical analyses are clearly detailed in the rebuttal letter, and it is important that this letter becomes part of the available files for this publication.